# Effects of Kiwifruit Dietary Fibers on Pasting Properties and In Vitro Starch Digestibility of Wheat Starch

**DOI:** 10.3390/nu16050749

**Published:** 2024-03-05

**Authors:** Yaqi Wang, Yaoyi Pan, Chang Zhou, Wenru Li, Kunli Wang

**Affiliations:** School of Nursing and Rehabilitation, Cheeloo College of Medicine, Shandong University, Jinan 250012, China

**Keywords:** kiwifruit dietary fibers, wheat starch, pasting properties, in vitro starch digestibility

## Abstract

In this study, the roles of kiwifruit soluble/insoluble dietary fiber (SDF/IDF, respectively) in the pasting characteristics and in vitro digestibility of wheat starch were explored. According to RVA and rheological tests, the IDF enhanced the wheat starch viscosity, decreased the gelatinization degree of the starch granules, and exacerbated starch retrogradation. The addition of SDF in high quantities could reduce the starch gelatinization level, lower the system viscosity, and exacerbate starch retrogradation. Through determining the leached amylose content and conducing scanning electron microscopy, the IDF and SDF added in high quantities was combined with the leached amylose wrapped around the starch granules, which reduced the leached amylose content and decreased the gelatinization degree of the starch granules. The Fourier transform infrared results showed that the addition of both the IDF and SDF resulted in an enhancement in hydrogen bonding formed by the hydroxyl groups of the system. The in vitro digestion results strongly suggested that both the IDF and SDF reduced the wheat starch digestibility. The above findings are instructive for the application of both IDF and SDF in starchy functional foods.

## 1. Introduction

Starch is a main plant storage carbohydrate and is a critical energy source for the human body [1]. Nonetheless, energy excess has become a social issue, as hyperglycemic responses caused by the chronic overconsumption of starchy foods have been associated with the development of cardiovascular diseases, diabetes, and other noncommunicable diseases [2]. In particular, diabetes, which is tightly related to glucose metabolic disorders, has become the major global issue threatening human and society health and well-being. Dietary interventions through modulating starch digestion have been proposed to prevent diabetes [3]. Based on starch use post-consumption, starch can be divided into three types, namely, rapidly digestible starch (RDS, digestion in a 20 min period), slowly digestible starch (SDS, digestion in a 20–120 min period), and resistant starch (RS, undigested after 120 min) [4]. It has been widely reported in the literature that RDS is a major cause of postprandial glycemic elevation, thus increasing the incidence rates of cardiovascular disease, type 2 diabetes, and obesity [5,6]. Additionally, RS has plenty of physiological effects, including the prevention of colon cancer, increased mineral absorption, reduced fat accumulation, and hypoglycemic and hypocholesterolemic effects [7]. A number of factors influence the starch fraction composition, such as the starch type and structure, food matrix processing, and exogenous compound entry in starchy foods [8,9].

Among them, dietary fiber, both endogenous and exogenous, plays an important role in reducing starch digestibility while lowering postprandial blood glucose levels, making its use a frequently adopted strategy in the processing of hypoglycemic-responsive foods [10,11]. Dietary fibers are generally classified as soluble or insoluble dietary fibers based on their water solubility. To date, studies have found that dietary fiber inhibits granule swelling and the leaching of amylose [12], increases the pasting temperature [13,14], inhibits the disintegration of starch granules [15], inhibits short- and long-term retrogradation [16], reduces starch digestibility [17], etc. Since most of these effects of dietary fibers on starch are positive, more and more novel dietary fibers are being added to starch to investigate their effects on starch properties.

Kiwifruit (*Actinidia deliciosa*) is a characteristic fruit planted in central and southern China and is a commercial fruit with worldwide importance [18,19]. The total, insoluble, and soluble dietary fiber (TDF, IDF, and SDF, respectively) levels of kiwifruit are 25.8%, 7.1, and 18.7% on a dry weight (dw) basis, respectively [19,20]. As a result, kiwifruit is a candidate for enriching dietary fiber products, increasing the commercial value of the fruit [21,22]. However, most of the current studies on kiwifruit dietary fiber have focused on the inhibition of digestive enzyme activity, hypolipidemic effects, hypoglycemic effects, and alterations in gut microbiota in relation to the nutritional value of kiwifruit dietary fiber [23,24], and there is limited scientific evidence supporting the pasting, structural, rheological, and textural characteristics of starch with added kiwifruit dietary fiber. Consequently, this work analyzed the pasting, rheological, and digestive characteristics of starch containing kiwifruit dietary fiber. The results shed light on the possible applications of kiwifruit dietary fiber in starchy foods and find new possibilities for the design of dietary-fiber-enriched starchy foods.

## 2. Materials and Methods

Wheat starch was provided by Yuanye Biotechnology Co., Ltd. (Shanghai, China). Spectral-grade KBr was provided by Aladdin Reagent Co., Ltd. (Shanghai, China). A leached amylose assay kit and 3,5-dinitrosalicylic acid (DNS) were provided by Beijing Solarbio Biotechnology Co., Ltd. (Beijing, China). Amyloglucosidase (300 AGU/g, AMG 300 L) was provided by Novozymes Biotechnology Co., Ltd. (Shanghai, China). Porcine pancreatic α-amylase (40.4 U/mg) was provided by Sigma-Aldrich (St. Louis, MO, USA). Kiwifruit SDF and IDF were isolated from fresh kiwifruit by our previously described research method [23].

### 2.1. Pasting Characteristics

The pasting characteristics of the wheat starch and starch/DFs mixtures were measured by a rapid visco analyzer (RVA; Newport Scientific, Warriewood, NSW, Australia). Mixtures of 5%, 10%, and 15% (*w*/*w*) IDF/wheat starch and 1%, 2%, and 3% (*w*/*w*) SDF/wheat starch were prepared in advance. The wheat starch or mixture (3.0 g) was added to 25.0 g of ultrapure water, stirred well, and stabilized at room temperature for 15 min before using RVA Standard 1 for measurements. The RVA system software (Thermocline for Windows (TCW3)) was utilized to calculate the peak viscosity (PV; maximum paste viscosity achieved in the heating stage of the profile), breakdown viscosity (BV; difference between peak and trough viscosities), trough viscosity (TV; minimum paste viscosity achieved after maintaining the mixtures at the maximum temperature), final viscosity (FV; viscosity at the end of the run), and setback viscosity (SV; difference between final viscosity and trough viscosity). The RVA program was employed for obtaining the sample RVA curve.

### 2.2. Fourier Transform Infrared (FT-IR) Spectroscopy

An FT-IR spectrometer (Spectrum 100, PerkinElmer Co., Shelton, CT, USA) was utilized to measure alterations in the functional groups in the wheat starch and wheat starch/DF mixtures. The starch gel obtained from the RVA experiment described in Section 2.1 was freeze-dried, mixed with dried spectral-grade KBr, and milled into a powder (2 mg sample/200 mg KBr). The milled powder was placed in a mold and pressed into thin sheets. The samples were scanned at wavelengths of 400–4000 cm^−1^.

### 2.3. Scanning Electron Microscopy (SEM)

First, we used a rapid visco analyzer (RVA; Newport Scientific, Warriewood, NSW, Australia) to gelatinize the samples, as described in Section 2.1. The gelatinized samples were then lyophilized for subsequent experiments. The micro-morphology of the samples were observed with a scanning electron microscope (SEM, SU8010, Hitachi, Ltd., Tokyo, Japan) following a previously described method [25] with mild modifications. After drying, the samples were subjected to fine grinding, and a portion of them was acquired onto a conductive adhesive followed by thin-layer gold coating. Sample observation was conducted at a 10.0 kV accelerating potential.

### 2.4. Rheological Measurements

The samples were gelatinized using a rapid visco analyzer (RVA, Newport Scientific, NSW, Warriewood, Australia) as described in Section 2.1 for subsequent experiments. A rheometer (DHR-2, TA Instruments, New Castle, DE, USA) was utilized to analyze the specific rheological characteristics using a 40 mm flat plate probe according to the method adapted from Zhang, Liu, Mo, Zhang, and Zheng [26]. The stage temperature was kept constant at 25 °C, and the spacing between the probe and the platform was set to 1000 μm.

Dynamic rheological measurements: The sample was placed on the plate of the rheometer and stabilized for 30 seconds. The sample was first subjected to a strain scan (1–100%) to determine the linear viscoelastic zone. A frequency scan (1–10 Hz) was then performed by selecting a 1% strain, based on the linear viscoelastic zone, to observe the changes in the sample’s storage modulus (G′), loss modulus (G″), and loss angle (tanδ = G″/G′).

Static rheological measurements: The sample was placed on the plate of the rheometer and stabilized for 30 s. A progressive increase in shear rate from 1 s^−1^–100 s^−1^ was observed, with a measurement duration of 90 s.

### 2.5. Leached Amylose

In this section, 1.0 g of wheat starch or a wheat starch/DF mixture containing an equal amount of starch was added to 20.0 g of water with 3 glass beads and subjected to a boiling water bath for 20 min for complete pasting. After cooling to room temperature, centrifugation was performed for 20 min at 4000× *g*, and 100 μL of the supernatant was taken to determine the content of leached amylose using a leached amylose assay kit (Beijing Solebo Biotechnology Co., Ltd., Beijing, China).

### 2.6. In Vitro Starch Digestion

The characteristics of starch digestion after pasting were determined based on the established methods [27,28]. A mixed enzyme solution was prepared by combining 250 U AMG glucosidase (300 U/mL) and 13,000 U porcine pancreatic α-amylase (40.4 U/mg), and it was then added to 25 mL of pH 5.2 acetate buffer.

A total of 1.0 g of wheat starch or a wheat starch/DF mixture containing an equal amount of starch was added to 100 mL of acetate buffer (pH 5.2), to which 3 glass beads were introduced, followed by 20 min of heating in a boiling water bath until complete pasting.

The sample was cooled to 37 °C, and 25 mL of the enzyme mixture was introduced, and it was then digested in a 37 °C oscillating water bath (110 rpm). The enzymatic reaction was terminated by taking 1 mL of the digestion solution at 0, 20, and 120 min and heating it for 5 min in a boiling water bath. The glucose level of the samples was determined by the DNS method [29]. The formulas for calculating the RDS, SDS, and RS contents within the wheat starch were as follows:$$RDS\;(\%)=\left(G_{20}-GF\right)\times 0.9\times 100/TS$$
$$SDS\;(\%)=\left(G_{120}-G_{20}\right)\times 0.9\times 100/TS$$
$$RS\;(\%)=\left[TS-\left(RDS+SDS\right)\right]\times 100/TS$$
in which *G*_20_ and *G*_120_ stand for the glucose levels generated by hydrolysis within 20 min and 120 min, respectively, *GF* is the free glucose level within the starch, while *TS* is the total starch dry weight.

### 2.7. Glucose Diffusion Inhibition Index

A 100 mmol/L glucose solution was prepared, and IDF and SDF samples were added separately to achieve a DF concentration of 1% (*w*/*v*). Next, 25 mL of the mixture was placed in a 3500 D dialysis bag and dialyzed against 200 mL of ultrapure water. The glucose concentrations in the dialysate were analyzed at 10, 20, 30, 60, 90, and 120 min using the DNS method [29].

### 2.8. Statistical Analysis

Each experiment was carried out three times, and the results were presented as mean ± standard deviation. Means were analyzed by analysis of variance (ANOVA) and Duncan’s multiple comparisons test using SPSS20.0, with *p* < 0.05 indicating statistical significance. Plots were made using Origin 2019.

## 3. Results

### 3.1. Pasting Properties

Table 1 and Figure 1 display the RVA curves and parameters for the wheat starch, starch/SDF, and starch/IDF combinations. At room temperature, the starch granules are insoluble in water; however, as the temperature rises to a certain level, starch swelling and dissolution into the water occur, causing the viscosity to increase. As seen in Figure 1, all of the samples had nearly identical viscosities at the start of the test. However, as the temperature increased, starch gelatinization began, and the viscosity profiles of the starch and starch/DF mixtures both significantly increased and revealed differences. The interaction of the wheat starch with the dietary fiber primarily occurred during the gelatinization process, as indicated by the significant variations between the wheat starch and the starch/DF mixes.

The PV, TV, and FV of the wheat starch increased with the addition of the IDF, and this phenomenon became gradually evident with the increase in the IDF concentration (Table 1). The BV of the wheat starch mixture with 5% IDF increased slightly compared to that of pure starch, and the BVs of the mixtures with 10% and 15% IDF decreased significantly compared to that of pure starch. In addition, the increase in the SV with the addition of the IDF proved that the IDF exacerbated starch retrogradation and, to some extent, facilitated the rearrangement of the gelatinized disordered starch granules. However, the viscosity results were different with the addition of the SDF. The PV, TV, and FV decreased with increasing the SDF concentration. The BV increased, and the overall gelatinization was enhanced by the addition of the SDF. The SV was significantly reduced by the addition of the SDF, proving that SDF can inhibit the aging of wheat starch during the cooling phase of the system.

### 3.2. FT-IR Spectroscopy

Figure 2 displays the FT-IR profiles for the starch/DF mixes and gelatinized wheat starch. The absence of new absorption peaks in the starch with the addition of the IDF and SDF, compared to pure wheat starch, suggested that the dietary fibers and wheat starch did not form covalent bonds.

The peaks between 3100–3500 cm^−1^ were related to the generation of hydrogen bonds by hydroxyl groups. The addition of both the IDF and SDF shifted the peak near 3441 cm^−1^ towards a lower wave number, proving that the addition of both dietary fibers strengthened the hydrogen bonding formed by the hydroxyl groups of the system [30]. The peak near 1640 cm^−1^ was associated with water absorption in the starch’s amorphous region. Adding both the SDF and IDF shifted this peak towards a lower wave number, and the addition of the IDF broadened the peak. This demonstrated that both SDF and IDF could enhance the binding of water to the system. In addition, the addition of the IDF also broadened and strengthened the peak at 2931 cm^−1^, proving that IDF strengthens the intermolecular hydrogen bonding of the system [31].

### 3.3. SEM

Figure 3 shows the microscopic morphology of the gelatinized wheat starch and starch/IDF mixtures. The wheat starch gel had a honeycomb reticular structure (Figure 3A). This was due to the evaporation of water from the amorphous zone of the starch after vacuum freeze-drying, which in turn formed widely distributed irregular pores [32]. Compared to pure wheat starch, the loose and porous mesh structure of the wheat starch/IDF gels appeared thinner and more porous as more IDF was added (Figure 3B–D). After adding the IDF, the loose and porous mesh structure of the wheat starch gel appeared thinner, and the pores became larger compared to the pure starch. Since the pores were left behind by the freeze-drying of moisture, the IDF restricted the diffusion and exudation of water and facilitated the formation of a mesh structure [31]. In addition, IDF itself is highly absorbent, and an increase in IDF addition could bind more water. Therefore, as the amount of IDF added increased, the IDF locked in more water, and the starch/IDF gel had more and larger pores. The addition of the SDF, on the other hand, made the samples look thick and firm because the SDF could cover the starch surface and fill the voids of the gelatinized starch [33]. It could also reduce cross-linking between starch granules.

### 3.4. Rheological Characteristics

#### 3.4.1. Dynamic Rheological Characteristics

Figure 4 shows the dynamic rheological characteristics of the gelatinized wheat starch and gelatinized wheat starch/DF mixtures. G′ and G″ stand for the sample’s elastic and viscous behaviors, respectively.

The G′ values of all the specimens were higher than the G″ values, indicating that all specimens were predominantly elastic and closer to the solid state than to the liquid state. The G′ values of the starch/IDF gels were all higher than that of the pure starch gels, but G′ decreased with the increase in IDF addition. This might be due to the fact that the IDF could absorb the bound water, and the water used by the starch for pasting in the system was reduced, thus reducing the degree of pasting and resulting in a weakening of the elastic characteristics. The G″ values of the starch/IDF gels were all higher than the G″ values of the pure starch gels and increased depending on the IDF addition. The change in G″ was significantly greater compared to G′, suggesting the greater impact of the IDF on the wheat starch viscosity [31]. The G′ values of the starch/SDF gels were lower than the G′ values of the pure starch gels, and all of them decreased with the increase in the amount of SDF added. This indicated that the SDF significantly reduced the elastic properties of the pasted starch and made the pasted starch closer to a liquid. In addition, the G″ of each starch/SDF gel decreased relative to pure starch, but this trend was not as pronounced as that of G′ with the increase in the amount of SDF added. Therefore, the SDF more significantly affected the elasticity of the pasted wheat starch, and SDF might suppress the expression of persistent wheat starch granule reticulation [34].

#### 3.4.2. Static Rheological Characteristics

Figure 5 displays the shear stress curves for the gelatinized wheat starch and starch/DF mixtures, while Table 2 presents the fitted parameters of the power law model. The R^2^ values were all greater than 0.989, suggesting a high model fitting degree for the shear stress and shear rate data.

The flow behavior indices (n) for all the specimens were less than 1, which indicated that the specimens exhibited pseudoplasticity [35]. The value of K represents the viscosity magnitude, considering consistency [36]. From Table 2, it can be seen that the K value showed a gradual increase as the IDF addition increased, suggesting that adding the IDF resulted in a greater resistance to flow [37]. However, the K value changed in the opposite direction after adding the SDF, indicating that there was shear thinning behavior [37]. Furthermore, it can be seen from Figure 5 that the higher the addition of IDF, the higher the required shear stress, while the opposite effect was observed with SDF on shear stress.

### 3.5. Amount of Leached Amylose

As the temperature increases, starch granule swelling within water begins, accompanied by the leaching of amylose [38]. The effects of the IDF and SDF on the amount of amylose leached from the wheat starch during the pasting process can be observed in Figure 6. As observed, the amount of leached amylose gradually decreased with the addition of the IDF, which indicated that the IDF could suppress amylose leaching. The SDF had no significant effect on the leaching of amylose at low additions (1–3%). However, the effects of higher concentrations of SDF (5% and 10%) on the amount of leached amylose were investigated. The results showed that the amounts of leached amylose were 13.24% and 11.15%, respectively, when SDF was added at 5% and 10%; these data are not shown in the figure. This indicated that the SDF had a significant inhibitory effect on the leaching of amylose when the additions were increased to 5%. The role of non-starch polysaccharides in inhibiting leached amylose has been identified previously [39,40]. This phenomenon might be due to the fact that IDF and SDF attach to leached amylose, thus surrounding the starch granule surface and ultimately inhibiting starch granule swelling and reducing the amount of leached straight-chain starch.

### 3.6. In Vitro Starch Digestion

The IDF and SDF significantly affected the RDS, SDS, and RS contents in the wheat starch (Table 3). With the addition of both the IDF and SDF, the RDS level declined significantly, while the RS content increased significantly, indicating that both the IDF and SDF decreased the wheat starch digestibility. In addition, the IDF apparently increased the SDS level, with a quantitative effect relationship. The SDF could also increase the content of SDS, but it was not related to the amount added.

### 3.7. Glucose Diffusion Inhibition Index Measurements

According to Table 4, the diffusion rate decreased with the increase in time, and the inhibition of glucose diffusion by the SDF was higher than that by the IDF. In addition to the binding of the DFs to the glucose itself, the viscosity of the solution increased after adding the DFs, which was also a factor affecting the diffusion of glucose. When the adsorption of the DFs reached saturation, the glucose diffusion was only impacted by the viscosity. These experimental results revealed that DFs have the potential to inhibit glucose diffusion and reduce absorption.

## 4. Discussion

Our experiments showed that the gelatinization, rheology, digestion, and other properties of the wheat starch changed after adding the IDF and SDF. The interaction between wheat starch and dietary fiber primarily occurred during the gelatinization process, as indicated by the considerable variations between the wheat starch and starch/DFs mixes.

During the pasting process, the PV and FV values increased after the addition of the IDF, probably due to the strong water-absorbing property and certain viscosity of IDF [31], which would compete with starch to absorb water in the continuous phase during heating, resulting in increased viscosity in the blended system [41,42]. The measurements of G″ in the rheological properties also suggested that adding the IDF dramatically affected the viscosity of the starch gels. The BV of the mixture with a 5% IDF addition slightly increased compared to pure starch, and the BV values of the mixtures with 10% and 15% IDF additions both showed a significant decrease compared to pure starch. This might be because the starch had less water to access and utilize after the addition of 10% and 15% IDF, resulting in a low degree of pasting and reduced solubility and therefore a lower BV value and increased starch stability. It was also possible that the IDF cross-linked with leached amylose and surrounded the starch granules, decreasing the starch granule pasting degree and also further suppressing amylose leaching. The lower BV led to a better resistance of the system to thermal and mechanical treatments [30]. In addition, the increased SV proved that the IDF exacerbated starch retrogradation and somewhat promoted the rearrangement of the pasted and disordered starch granules. This result might be associated with the IDF-bound water, which decreased starch chain mobility [43].

The addition of the SDF reduced the starch gel system’s viscosity. This result was consistent with previous studies reported for non-starch polysaccharides [14,44,45], which is probably because the SDF adhered to leached amylose before surrounding the starch granule surface, reducing the cross-linking degree of the starch granules and lowering the system viscosity [45]. This conjecture was further confirmed by the results of the SEM experiments in this study (Figure 3E–G). The BV became larger, and the overall pasting was enhanced by the addition of the SDF, but it weakened with the increase in the amount of SDF added. These results suggested that at a sufficiently high SDF content, the SDF could bind to the leached amylose during the pasting process and attach to the starch surface, thereby inhibiting starch granule swelling and fragmentation [28].

According to the in vitro digestion experimental analysis, adding the IDF and SDF reduced the RDS level while increasing the SDS content, revealing that both the IDF and SDF decreased the wheat starch digestibility. The reasons that the IDF could decrease the wheat starch’s in vitro digestibility were as follows: (1) based on the results of the RVA, it can be seen that the IDF reduced the starch granule gelatinization degree, while the digestion of unfragmented starch was difficult, causing the decreased RDS level; (2) as shown by the results of the FT-IR spectroscopy, the IDF and leached amylose were bonded to each other by hydrogen bonding, which encircled the starch granule surface and reduced the opportunities for digestive enzymes to be able to hydrolyze the starch, promoting the elevated SDS and RS levels. The reasons that the SDF could decrease the wheat starch in vitro digestibility were as follows: (1) as shown by the results of the SEM, the SDF attached onto the starch surface, slowing down the starch decomposition rate by digestive enzymes, which led to a low RDS; (2) as shown by the results of the FT-IR spectroscopy, a small amount of SDF was bound to the leached amylose through hydrogen bonding, attached to the starch granules’ surface, and inhibited their swelling and fragmentation, which promoted the increase in RS. Combined with the inhibition of both the IDF and SDF against glucose diffusion, it can be further hypothesized that the inhibition of starch gelatinization by the IDF and SDF, as well as their physical barrier effects [31], contributed to the slowing down of glucose release during starch digestion and consequently to the reduction in postprandial blood glucose and insulin levels [46]. The above findings were consistent with those of *Mesona chinensis polysaccharides* (MCP) mixed with wheat starch, where the presence of MCP reduced starch digestion in gel fragments [47]. The monosaccharide composition, molecular weight, and structure of dietary fiber from different sources are different, and the gelatinization and digestion properties of starch added with various dietary fibers are also different. Therefore, the effects of different sources of dietary fiber on starch gelatinization and digestion need to be determined experimentally.

## 5. Conclusions

The functions of both IDF and SDF in starch pasting and in vitro digestibility were significant, and they were largely associated with their additive amounts. The addition of IDF and SDF decreased the RDS content while increasing the SDS level within wheat starch, indicating that both the IDF and SDF decreased the wheat starch digestibility, lowered the glucose release rate, and reduced the amount of glucose release. This study provided a basis for the application of IDF and SDF in wheat starch foods. Regarding the diverse biological effects of IDF and SDF, combining IDF and SDF with wheat starch can confer these functions to wheat starch for the development of functional foods, which requires further studies.

## Figures and Tables

**Figure 1 nutrients-16-00749-f001:**
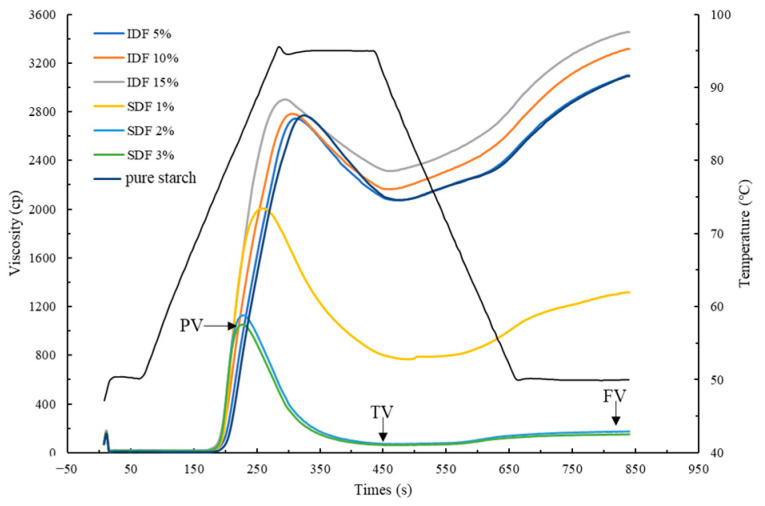
RVA curves showing wheat starch and starch/DF mixtures under diverse DF concentrations.

**Figure 2 nutrients-16-00749-f002:**
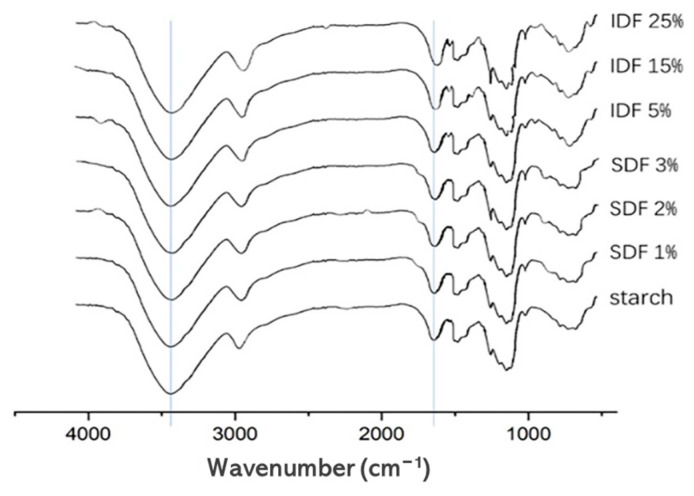
FT-IR spectra for lyophilized powder of wheat starch and starch/DF mixtures under diverse DF concentrations.

**Figure 3 nutrients-16-00749-f003:**
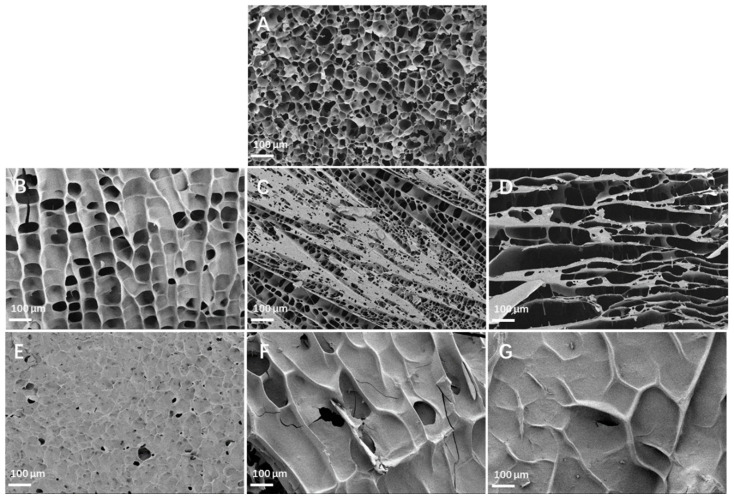
SEM images for lyophilized powder of wheat starch and starch/DF mixtures under diverse DF concentrations (100×): (**A**) starch; (**B**) IDF 5%; (**C**) IDF 10%; (**D**) IDF 15%; (**E**) SDF 1%; (**F**) SDF 2%; (**G**) SDF 3%.

**Figure 4 nutrients-16-00749-f004:**
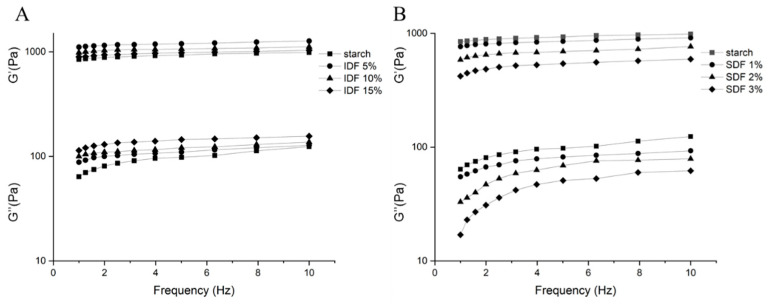
Storage modulus (G′) and loss modulus (G″) of gelatinized wheat starch and starch/DF mixtures at different DF concentrations: (**A**) IDF; (**B**) SDF.

**Figure 5 nutrients-16-00749-f005:**
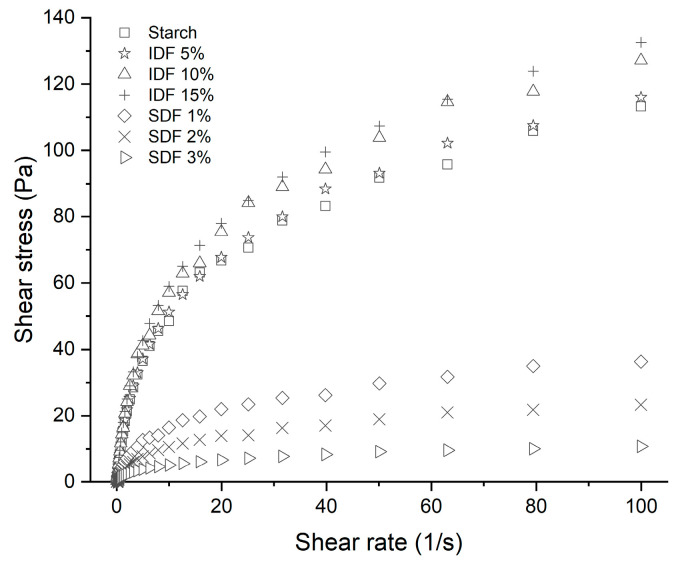
Steady flow curves for different gelatinized wheat starch and starch/DF mixtures.

**Figure 6 nutrients-16-00749-f006:**
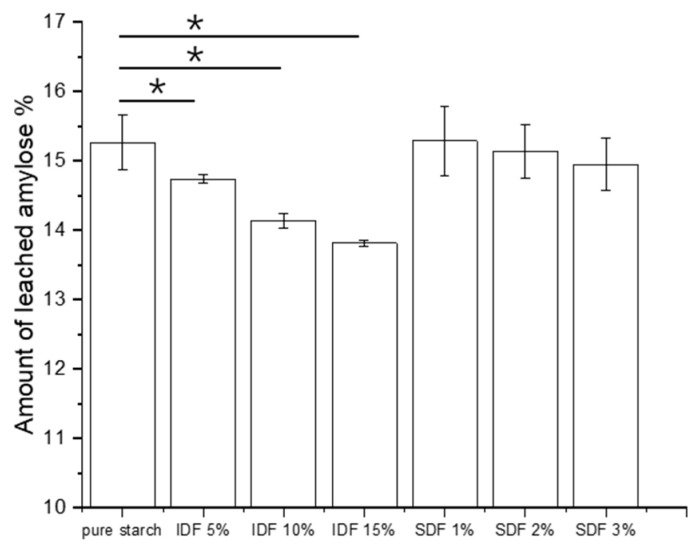
Leached amylose of wheat starch and starch/DF mixtures under diverse DF concentrations. Data are presented as mean ± standard deviation, * *p* < 0.05.

**Table 1 nutrients-16-00749-t001:** Viscosity characteristics of wheat starch and starch/DF mixtures under diverse DF concentrations.

	PV (mPa·s)	TV (mPa·s)	BV (mPa·s)	FV (mPa·s)	SV (mPa·s)
Wheat starch	2781.00 ± 21 d	2096.33 ± 31.77 c	684.67 ± 11.68 b	3096.33 ± 22.19 c	1000.00 ± 14.73 c
IDF 5%	2764.67 ± 19.76 d	2068.33 ± 18.45 c	696.33 ± 20.11 b	3112.00 ± 17.35 c	1043.67 ± 14.01 cd
IDF 10%	2857.33 ± 32.59 de	2218.33 ± 43.68 d	639.00 ± 11.14 ab	3395.33 ± 80.01 d	1177.00 ± 101.06 d
IDF 15%	2979.67 ± 24.85 e	2383.33 ± 8.39 e	596.33 ± 33.23 a	3518.33 ± 61.70 d	1135.00 ± 69.16 cd
SDF 1%	2004.33 ± 36.07 c	758.00 ± 37.64 b	1246.33 ± 4.73 e	1308.67 ± 60.93 b	550.67 ± 23.50 b
SDF 2%	1307.33 ± 157.81 b	138.00 ± 65.78 a	1169.33 ± 93.38 d	308.00 ± 124.07 a	170.00 ± 59.15 a
SDF 3%	980.67 ± 60.05 a	53.67 ± 9.07 a	927.00 ± 51.10 c	126.00 ± 22.72 a	72.33 ± 13.65 a

Diverse lowercase letters within one column suggest that the data are of significant difference (*p* < 0.05).

**Table 2 nutrients-16-00749-t002:** Rheological parameters of different gelatinized wheat starch and starch/DF mixtures.

	K (Pa sn)	n (-)	R^2^
Wheat starch	16.914 ± 0.096 d	0.429 ± 0.001 d	0.991
IDF 5%	17.155 ± 0.045 e	0.430 ± 0.001 de	0.99
IDF 10%	19.052 ± 0.015 f	0.432 ± 0.001 ef	0.989
IDF 15%	19.700 ± 0.021 g	0.432 ± 0.001 f	0.991
SDF 1%	5.634 ± 0.052 c	0.421 ± 0.001 c	0.992
SDF 2%	3.618 ± 0.035 b	0.418 ± 0.001 b	0.99
SDF 3%	1.798 ± 0.010 a	0.407 ± 0.001 a	0.992

Different lowercase letters in one column indicate that data in this column are of significant difference (*p* < 0.05).

**Table 3 nutrients-16-00749-t003:** The RDS, SDS, and RS contents in wheat starch and starch/CPS mixtures.

	RDS%	SDS%	RS%
Wheat starch	83.15 ± 2.20 d	10.93 ± 0.65 a	5.32 ± 2.05 a
IDF 5%	71.49 ± 2.23 c	17.81 ± 1.85 b	10.1 ± 4.07 ab
IDF 10%	63.51 ± 1.27 b	23.13 ± 1.77 cd	12.76 ± 1.90 bc
IDF 15%	57.06 ± 3.28 a	25.89 ± 3.34 d	16.45 ± 0.12 c
SDF 1%	67.34 ± 1.10 bc	18.87 ± 1.20 bc	13.19 ± 0.42 bc
SDF 2%	63.2 ± 0.64 b	18.32 ± 0.82 bc	17.89 ± 1.12 c
SDF 3%	56.7 ± 1.71 a	16.29 ± 1.79 b	26.42 ± 0.43 d

Different lowercase letters in one column indicate that data in this column are of significant difference (*p* < 0.05).

**Table 4 nutrients-16-00749-t004:** Effect of DF samples on dialysis of glucose.

Time (min)	10	20	30	60	90	120
Control	185.32 ± 2.98 c	484.16 ± 9.37 c	673.97 ± 17.02 c	851.89 ± 11.56 c	969.15 ± 15.23 c	1025.26 ± 5.50 c
SDF	127.66 ± 1.38 a	384.84 ± 3.12 a	526.04 ± 13.31 a	664.88 ± 5.44 a	780.94 ± 9.45 a	832.61 ± 13.57 a
IDF	160.11 ± 6.79 b	436.86 ± 11.07 b	593.99 ± 13.18 b	755.82 ± 6.90 b	887.48 ± 11.04 b	955.95 ± 28.73 b

Different lowercase letters in one row suggest significant difference in data in this row (*p* < 0.05).

## Data Availability

All data that support the findings of this study are available from the corresponding author on reasonable request.

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
