# Peer review of "Effects of Kiwifruit Dietary Fibers on Pasting Properties and In Vitro Starch Digestibility of Wheat Starch"

_nutrients, 2024, doi:10.3390/nu16050749_

Round 1
Reviewer 1 Report
Comments and Suggestions for Authors
Title: Effects of Kiwifruit dietary fibers on pasting properties and in vitro starch digestibility of wheat starch
The paper deals with the effect of soluble and insoluble kiwi fibers added to wheat starch, in order to evaluate its pasting properties and its ability to lower postprandial blood glucose response.
However, the work should be written better, there are content to be clarified, both in materials and methods and in the results.
The discussion is not very explanatory of the results and should be increased.
Line 93-94 and 100: What it means that starch gel sample were acquired from RVA. Can it be explained better?
Pasting properties:
Could you give a brief explanation of the parameters that you consider, also highlighting them in the graph?
Line 170 171: why gelatinization was enhanced but BV weakened?
In the paragraph relating to FT-IR spectroscopy it is not possible to understand, by the figure, what the peaks and the peak shift refer to. Can the figure be made clearer, perhaps by overlap the curves?
Amount of leaked amylose:
in this paragraph you spoke about samples with 5 and 10% soluble fibers which have never been mentioned. Either you include them in materials and methods but explain why you consider them only for this analysis, or you have to eliminate them.
Discussion: 313-317 the BV increased with 5% IDF but decreased with 10 and 15% but the discussion in line 315 is referred to at 5% or 10 and 15% of IDF?
On line 354: What is MCP?
Reviewer 2 Report
Comments and Suggestions for Authors
The manuscript “Effects of Kiwifruit dietary fibers on pasting properties and in vitro starch digestibility of wheat starch” describes the effect of dietary fibers from kiwifruit when added at different concentrations to wheat starch. The purpose is to decrease the negative effects of starch on blood glucose levels. This is a very interesting study where the authors have performed a number of different experiments to characterize the results of adding dietary fibers to starch. The results are very well described and the images are of high quality. The statistical analysis is correctly performed and the pertinent literature cited. However, there are some concerns with the manuscript:
Major concerns:
1: Figure 6: Please do not use letters to describe statistical analysis. It is impossible to understand what was compared to what. Use the conventional way of describing statistical significance, but adding a line between the two groups compared and stars above the line to indicate significance level. The same for the tables: It is impossible to understand what was compared to what. Please indicate significance with stars and describe what was compared to what with words, not letters.
2: Conclusion: This is the only sentence that should be in the conclusion section: “the addition of IDF and SDF decreased RDS content while increasing SDS level within wheat starch, indicating that both IDF and SDF decreased wheat starch digestibility, lowered glucose release rate, and reduced the amount of glucose release”. The rest is discussion. Please move to the discussion section.
3: In the discussion section, please discuss if the findings of the manuscript are specific to kiwifruit dietary fibers or if the same results could be reached with fibers from other sources.
Minor concerns:
Please revise the language to be proper English.
Comments on the Quality of English LanguageThe language is perfectly understandable, but needs some editing to sound like proper English.
